# End-Effectors Developed for Citrus and Other Spherical Crops

**Xu Xiao** [1,2], **Yaonan Wang** [1,2,*] and **Yiming Jiang** [1,2]

1    College of Electrical and Information Engineering, Hunan University, Changsha 410082, China
2    National Engineering Laboratory for Robot Visual Perception and Control Technology, Hunan University, Changsha 410082, China
*    Correspondence: yaonan@hnu.edu.cn

**Abstract:** Citrus harvesting is time-intensive and labor-intensive, relying mainly on manual harvesting. The automatic harvesting of fruit and vegetable crops can not only reduce the physical labor of fruit farmers in the harsh field environment but also greatly improve the harvesting efficiency. Based on the principle of manual citrus picking, an end-effector with three-finger grasping is designed in this study. First, the structure of the end-effector was designed to achieve the function of stable grasping and effective cutting of citrus fruits, and then the working process and key parameters of the end-effector were explained in detail. Finally, a picking test was conducted without considering robot vision. The test results show that the end-effector has a picking success rate of 95.23% for citrus with a diameter of 30–100 mm and an average picking time of 4.65 s for a single fruit. This end-effector can realize the picking function for citrus of different sizes and shapes and has the advantages of high adaptability, stable gripping and no damage to the fruit.

**Keywords:** picking trials; end-effector; citrus; robotic harvesting





## 1. Introduction

Citrus is one of the most popular fruits in China, with an area of 2788.6 thousand hectares and a production of 51.129 million tons by 2021, ranked first in the world [1]. Harvesting is one of the most important aspects of citrus production, which is seasonal and labor-intensive [2,3], and the cost of harvesting citrus manually accounts for 55% of the total cost of citrus production [4,5]. The citrus harvesting time is short, manual work is inefficient, and there is a risk of being cut by branches during operation [6]. However, mechanized harvesting can easily damage citrus fruits and thus fail to meet the market requirements [7]. In recent years, most of the young agricultural workers in China have turned to other industries, resulting in a shortage of agricultural labor, which has seriously affected the development of the fruit and vegetable industry [8]. Therefore, the development of highly intelligent picking robots has become a popular research topic in today's society. Picking robots can not only free labor, reduce production costs and improve operational efficiency but also improve the quality of citrus picking, meet the demand of citrus production in real-time and ensure the freshness of the fruit [9].

In recent years, scholars at home and abroad have conducted research on end-effectors for fruit and vegetable picking, and different end-effectors have been developed for different kinds of crops such as tomato [10,11], strawberry [12], apple [13] and kiwifruit [14]. For citrus picking, the end-effector needs to adapt to different sizes and ellipticities of citrus picking needs, and it needs to complete the separation between citrus fruits and stems [15] to achieve stable picking, which is one of the key technologies for the research of end-effectors of picking robots.

To solve the problem of nondestructive citrus picking, most scholars have used soft materials and flexible actuators [16]. However, due to the inherent instability of the pneumatic system, the problem of fruit dropping due to unstable grasping may occur. Citrus grown in the natural environment can have large differences in size and ellipticity [17,18]. While the

traditional finger-clamp end-effector mechanism is simple and can effectively grasp the fruit by clamping the fingers, the fruit stalk separation is prone to cause peel breakage of the fruit [19,20].

For example, the 10-degree-of-freedom under-driven three-finger hand [21–30] developed by Gosselin's group at Laval University in Canada, which is the international leader in the under-driven multi-finger hand control, uses only two motors, one motor is responsible for the grasping opening and closing motion of three fingers, and the other motor completes the finger steering. Rodriguze et al., developed a 15-degree-of-freedom single-motor-driven multi-finger hand, which can achieve safe and reliable grasping without any sensors and feedback control [31]. At present, most multi-finger hands with bending and torsional complex degrees of freedom and active soft control are used for humanoid dexterous operation, integrating multiple sensors, motors and actuators, with complex control and high cost [32–35].

In order to provide a picking robot with a simple and inexpensive structure, with a certain degree of flexibility, and that is suitable for a variety of sizes of citrus and other spherical picking end-effectors, the actuator uses a finger bending grasping mechanism to achieve the grasping of different sizes of spherical fruits, such as citrus, and to rotate the cutting between the blade and the stalk. We make a prototype for experimental verification. The focus of this study is on the picking mechanism. The machine's vision system is not in the scope of the study.

## 2. Materials and Methods

### 2.1. Overall Design of the End-Effector

By analyzing the picking of citrus by human hands, the movement of human hands can be divided into two types: stable grasping by fitting the inner side of the fingers to the surface of citrus and cutting of fruit stems by a blade. Considering that the task of picking can be completed, and the structure needs to be simple and easy to control, it was decided to design the end mechanism as a three-finger structure to realize the two grasping methods. The gripping process is when the end-effector reaches the designated position, and the motor starts and drives the fingers to perform the gripping action through the mechanical structure. When the gripping force reaches the rated value, the finger stops moving and completes the gripping action; then, the base starts to drive the citrus to rotate so that the blade between the two fingers completes the cutting of the citrus fruit stalk and separates the citrus from the fruit stalk.

As shown in Figure 1, the designed three-finger clamping end-effector consists of; 1. a connecting chuck, 2. fixed bracket, 3. clamping fingertip, 4. moving linkage, 5. stepping motor, 6. blade-fixed frame, 7. cutting blade, 8. clamping finger. The end-effector can be fixed at the end of the robot arm to complete the action of clamping the citrus fruit and cutting the fruit stalk to realize the picking operation of different kinds of citrus fruit.

The end-effector of the picking robot holds the citrus fruit with its clamping fingers. To adapt to the different sizes of citrus fruits, the clamping fingers are driven by the stepper motor to maximize the opening of the clamping fingers before grabbing the fruits. When the end-effector reaches the picking point, the gripping finger slides down under the positive drive of the stepper motor, thus driving the gripping finger to close the movement until the gripping finger makes adaptive contact with the target fruit to form a certain envelope on the surface of the citrus, generating a large friction force with a small gripping force to reliably grab the citrus fruit without causing damage to the citrus fruit. After the gripping fingers grip the citrus fruit, the end-effector rotates as a whole so that the cutting blade fixed between the gripping fingers finishes cutting the citrus fruit's stalk.

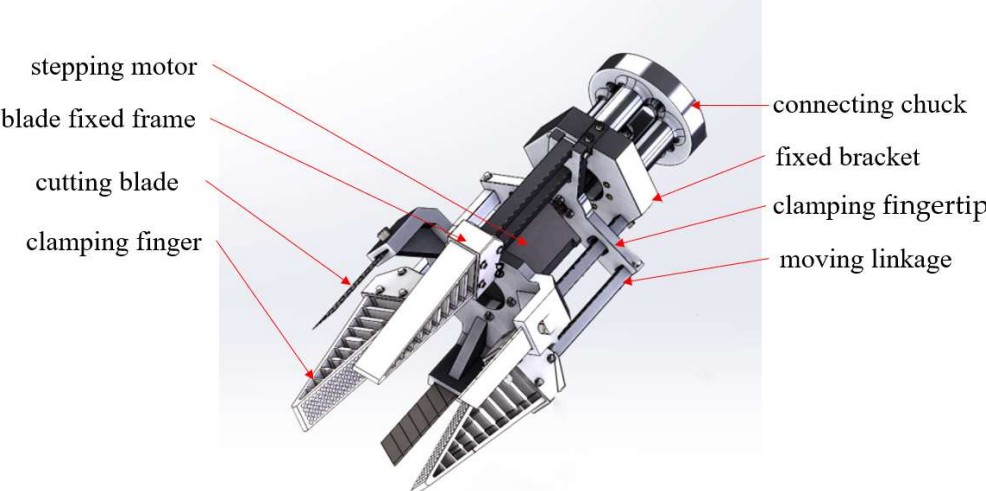

**Figure 1.** Structure diagram of the end-effector.

### 2.2. Clamping Mechanism Design

2.2.1. Clamping Finger Form

The citrus picking end-effector is first fixed by clamping citrus fruits. The gripping finger is the part of the end-effector that is in direct contact with the citrus fruit, so the material of the finger will have a certain influence on the picking effect of the citrus fruit. The end-effector developed by Monta et al. [36–39] in Japan uses a flexible finger driven by multiple joints, which can be adaptable to the size difference of fruits. However, there is serious shading in the re-growth of citrus fruits, and citrus fruit trees have more lush branches and often grow multiple fruits on one branch. The citrus picking robot is often affected by the branches and the adjacent fruit in the picking process, and the end-effector bends when picking the target fruit and causes damage to the target fruit. The fin-like flexible finger can better adapt to the size of citrus fruit, it will not damage the surface of the skin of citrus during the picking process, and it can be 3D printed with ABS material at low cost. Therefore, this paper selects the fin-like flexible finger as the clamping finger of the picking end-effector.

2.2.2. Number of Fingers

According to the spherical structure of citrus fruits, two or more gripping fingers are generally used for gripping [40]. However, two clamping fingers are unstable for the spherical structure of the fruit, and it is necessary to ensure that the clamping fingers are clamped near the equatorial surface of the citrus in order to make the clamping of the fruit stable, but there are often visual positioning errors and mechanical arm control errors when the picking robot is working on picking, etc. Multiple clamping fingers for fruit grasping will lead to a more complex end-effector control system and reduce the end-effector's picking efficiency, increasing the cost [41]. In summary, considering the problems of picking accuracy and cost, this paper adopts a three-finger end-effector, which can effectively ensure the error of the picking robot in the working process and guarantee the effectiveness of the control system.

2.2.3. Mathematical Model of Citrus Picking Finger Length

Due to the great differences in the growth characteristics of citrus fruits, in order to determine the effective grip of the picking finger on the target fruit, according to the measurement results of the physical characteristics of the actual citrus fruit, it is known that the horizontal diameter of mature citrus fruit is 48.25–80.85 mm, the average citrus fruit horizontal diameter is 62.64 mm; the vertical diameter range is 41.83–72.62 mm, the average citrus fruit vertical diameter is 55.85 mm; the diameter range of citrus fruit stalks is

1.25 to 3.03 mm, with an average citrus fruit stalk diameter of 2.03 mm. Different diameters of citrus-picking schematic citrus fruit can be defined as spheres [42]. When the gripping fingers of the end-effector have gripped and wrapped the citrus fruit, the large diameter fruit was gripped as shown in Figure 2a, and the small diameter fruit was gripped as shown in Figure 2b. In order to determine the length of the clamping finger, a clamping mechanism model with the equatorial plane of the citrus fruit as the base plane was established with the range of the citrus fruit's transverse diameter as the index. In order to ensure the effective clamping of the clamping finger on the citrus fruit, the clamping coverage area of the clamping finger must be greater than 55% when picking large diameter fruit. According to the analysis in Figure 2, the length of the clamping finger is 70 mm.

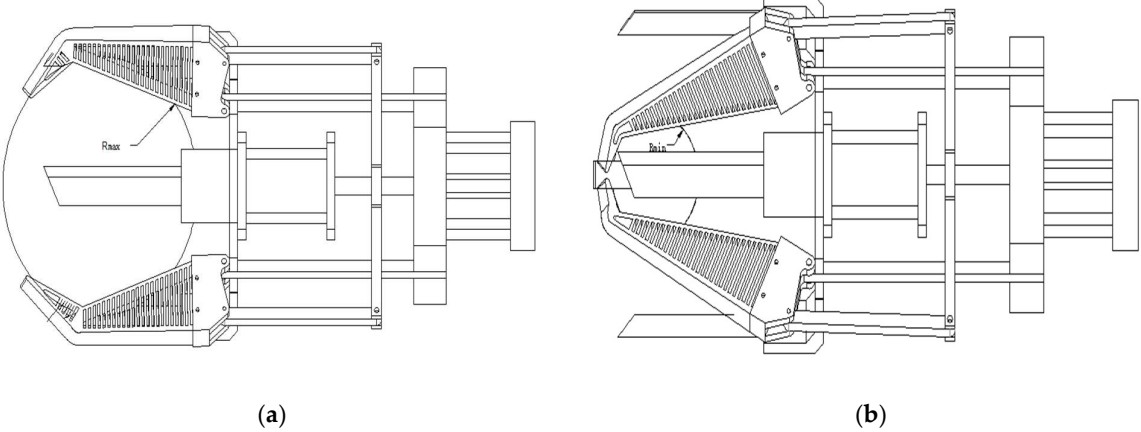

(**a**)                (**b**)

**Figure 2.** Diagram of citrus picking at different sizes and diameters. (**a**) Large-diameter citrus picking map; (**b**) Small-diameter citrus picking map.

### 2.2.4. Analysis of Stresses on Citrus Fruits

When the end-effector carries out the picking operation, the first gripping finger will grasp the fruit. There are six force points between the three flexible fingers and the citrus skin; the mathematical model of citrus and its force analysis is shown in Figure 3.

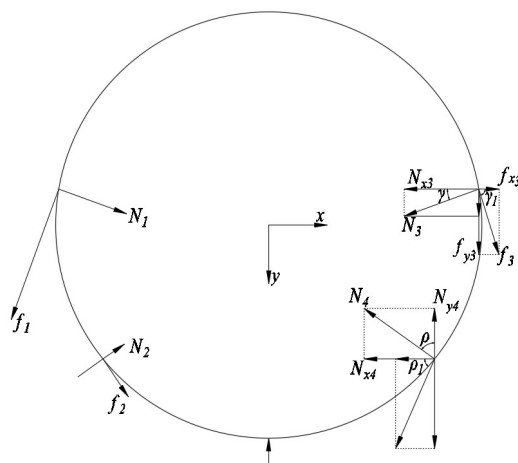

**Figure 3.** Schematic diagram of the forces on citrus. Note: $f_i$ is the static friction force in the vertical direction at the i-th contact point, N; $N_i$ is the positive pressure at the ith contact point, N; i = 1,2,3,4; mg is the gravity of the citrus fruit, N.

The contact type between citrus and gripping fingers can be considered as point contact with friction, and when the gripping fingers tighten on the citrus fruit at this time, the critical condition that the citrus can be held stably without slipping is that the gravity

of the fruit is equal to the maximum static friction between the fruit and the fingers [43]. The pressure between the citrus skin and the end-effector is set to N. Since the end-effector finger is centrally symmetric, the pressure between the clamping finger and the citrus skin is equal, i.e.,

$$N_1 = N_2 = N_3 = N_4 = N_5 = N_6 \qquad (1)$$

When the end-effector gripping finger holds the citrus steadily, the force analysis is as follows:

$$
\begin{aligned}
\sum_{i=1}^{6} f_i &= mg \\
N_{\min} &= \min\left(\frac{f_1}{\mu}, \frac{f_2}{\mu}, \frac{f_3}{\mu}, \frac{f_4}{\mu}, \frac{f_5}{\mu}, \frac{f_6}{\mu}\right) \\
N_{\min} &\geq \frac{m_{\max}g}{6\mu} \\
f_i &= \mu N_i
\end{aligned}
\qquad (2)
$$

where $m_{\max}$ is the maximum mass of the citrus fruit, kg;

$g$ is the acceleration of gravity, taken as 9.8 m/s$^2$;

$\mu$ is the coefficient of friction between the citrus skin and the flexible finger contact material.

Applying a certain cushioning material on the surface of the pressure sensor on the three gripping fingers [44] can increase the friction coefficient and improve the surface softness of the fingers, which, in turn, can reduce the required output positive pressure an, at the same time, can reduce the mechanical damage that may occur during the gripping process, which is conducive to achieving a soft grip. Different materials for soft gripping fingers can cause different damage to citrus. To determine the best cushioning material, the friction coefficient between the citrus peel and ABS material was 0.64 based on the friction characteristics of citrus peel in Section 2 and considering that the biological yield point of citrus is located between 10 and 20 mm of displacement [45], and the rupture point is located about 10 mm after the biological yield point. Therefore, ABS (3D printing material) was chosen as the flexible gripping finger material.

### 2.2.5. Clamping Mechanism Parameter Setting

Static analysis can be representative because of the low speed of the fingers when they work [46,47]. Since the three clamping fingers and the crank slider mechanism are symmetrically arranged along the center of the ball screw, one side is taken for analysis, and the clamping fingers of the end-effector are composed of a four-link drive mechanism, which can be known according to the vector analysis method in the mechanical principles. Then, the rewriting into a complex vector form is:

$$l_1 e^{i\theta_1} + l_2 e^{i\theta_2} + l_3 e^{i\theta_3} - l_4 = 0. \qquad (3)$$

Substitution according to Euler's formula yields:

$$l_1\cos\theta_1 + l_2\cos\theta_2 + l_3\cos\theta_3 - l_4 = 0 l_1\sin\theta_1 - l_2\sin\theta_2 - l_3\sin\theta_3 = 0. \qquad (4)$$

where $\theta_1$ and $\theta_2$ are solvable.

Relating the above equation yields:

$$(l_4 - l_3\cos\theta_3 - l_1\cos\theta_1)^2 + (l_1\sin\theta_1 - l_3\sin\theta_3)^2 = l_2^2 \qquad (5)$$

The solution gives:

$$\theta_3 = \varphi - \frac{\arcsin\left(l_4^2 + l_1^2 + l_3^2 - l_2^2 - 2l_4 l_1\cos\theta_1\right)}{2l_3\sqrt{l_4^2 + l_1^2 - 2l_4 l_1\cos\theta_1}} \qquad (6)$$

Among them.

$$\varphi = \frac{\arctan(l_4 - l_1\cos\theta_1)}{l_1\sin\theta_1} \tag{7}$$

It can be concluded that:

$$\theta_3 = \frac{l_1[l_4\sin\theta_1 l_3\sin(\theta_1 + \theta_3)]}{l_3[l_4\sin\theta_3 l_1\sin(\theta_1 - \theta_3)]}\theta_1 \tag{8}$$

When the end-effector grips the finger to wrap the citrus, as shown in Figure 4, the change in the gripping finger joint before contacting the fruit is the same as the precise gripping, and when the first end joint stops rotating, the end knuckle contacting the fruit begins to rotate relative to the joint, which can yield the change in angle between the rods [48,49].

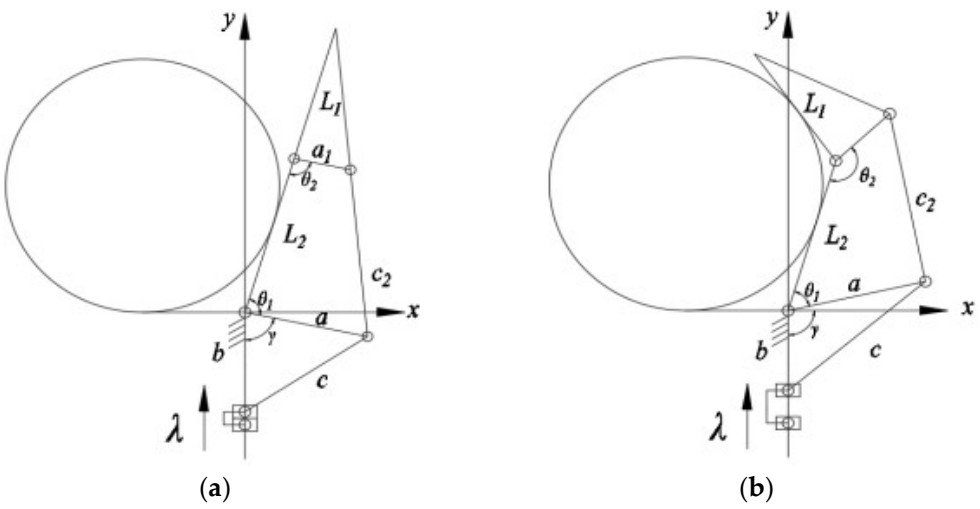

**Figure 4.** Sketch of clamping fingers holding fruit. (**a**) Force analysis before clamping citrus fruits (**b**) Force analysis when clamping citrus fruits.

$$\theta_1 = \frac{3\pi}{2} - \lambda - \phi \tag{9}$$

Among them,

$$\gamma = \arccos\frac{a^2 + (b - \lambda^2) - c^2}{2a(b-\lambda)},$$
$$\theta_2 = \varphi_1 - \arcsin\frac{l^2 + b_1^2 + a^2 - c_1^2 - 2lb_1\cos\theta_1}{2a\sqrt{l^2 + b_1^2 - 2lb_1\cos\theta_1}} \tag{10}$$
$$\varphi_1 = \frac{\arctan(l - b_1\cos\theta_1)}{b_1\sin\theta_1}$$

### 2.2.6. Control System Design

The stepper motor controller uses STM32F103C816, adding a 12 V power supply module to provide the required power for the stepper motor. The power of the motor is relatively large in order to protect the control circuit and drive circuit; so that the stepper motor does not burn the other components in the event of a short circuit and other faults; the circuit is added to the optocoupler isolation. We add an A4988 compiler to the circuit to adjust the motor's steering and motor steps, a serial communication module to realize data transmission between the control board and the upper computer, and key control in the circuit: KEY0 controls the rotation and stop of robot arm 1; KEY1 controls the positive rotation of robot arm 1; KEY2 controls the reverse rotation of robot arm 1; KEY3 controls the reverse rotation of robot arm 2; KEY4 controls the positive rotation of robot arm 2; KEY5 controls the rotation and stop of robot arm 2; SW1 is the reset button. When the picking operation starts, the switch is opened to provide 12 V current for the stepper motor, and

the current is shunted to the microcontroller for power supply when it passes through A4988 so that the microcontroller starts the stepper motor; the optocoupler isolates and uploads the signal to the stepper motor to control the lateral movement of the stepper motor. When the stepper motor moves up, it pushes the linkage mechanism to control the gripping finger for the opening movement; when the stepper motor moves down, it pushes the linkage mechanism to control the gripping finger for the closing movement. The schematic diagram of the control system is shown in Figure 5.

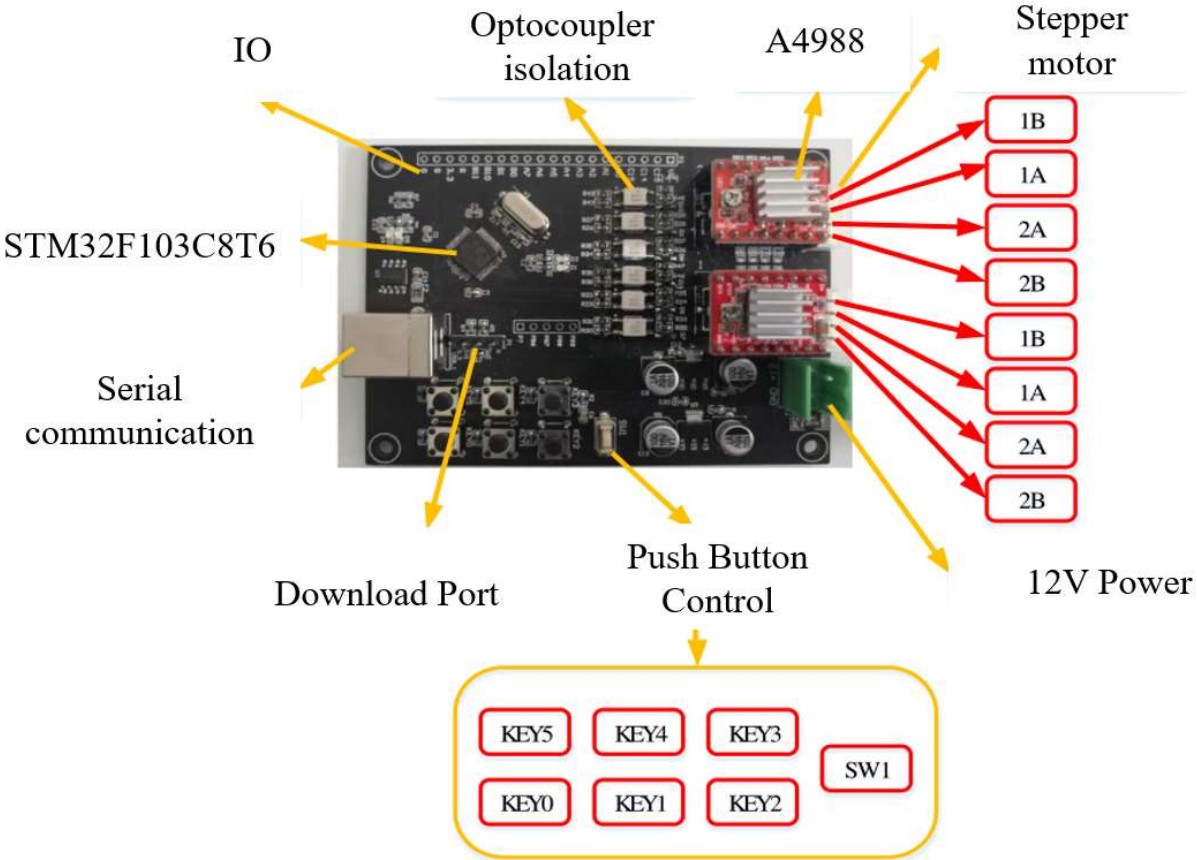

**Figure 5.** Schematic diagram of the control system structure.

　　　The gripping process of this end-effector system consists of two parts: finger bending and overall rotation of the end-effector, as shown in Figure 2. The actuator finger bending has good shape adaptiveness; when grasping larger size citrus fruit, each joint finger surface can fit; when grasping smaller size citrus fruit, it can complete the parallel pinching of fingertips. The end-effector's overall rotation is based on feedback from the active soft control when grasping different elliptical citrus according to the motor current feedback drive joint rotation, the axial rotation force coupling is eliminated, and the finger surface perpendicular to the cross-section tangent. The end-effector's picking operation flow is shown in Figure 6.

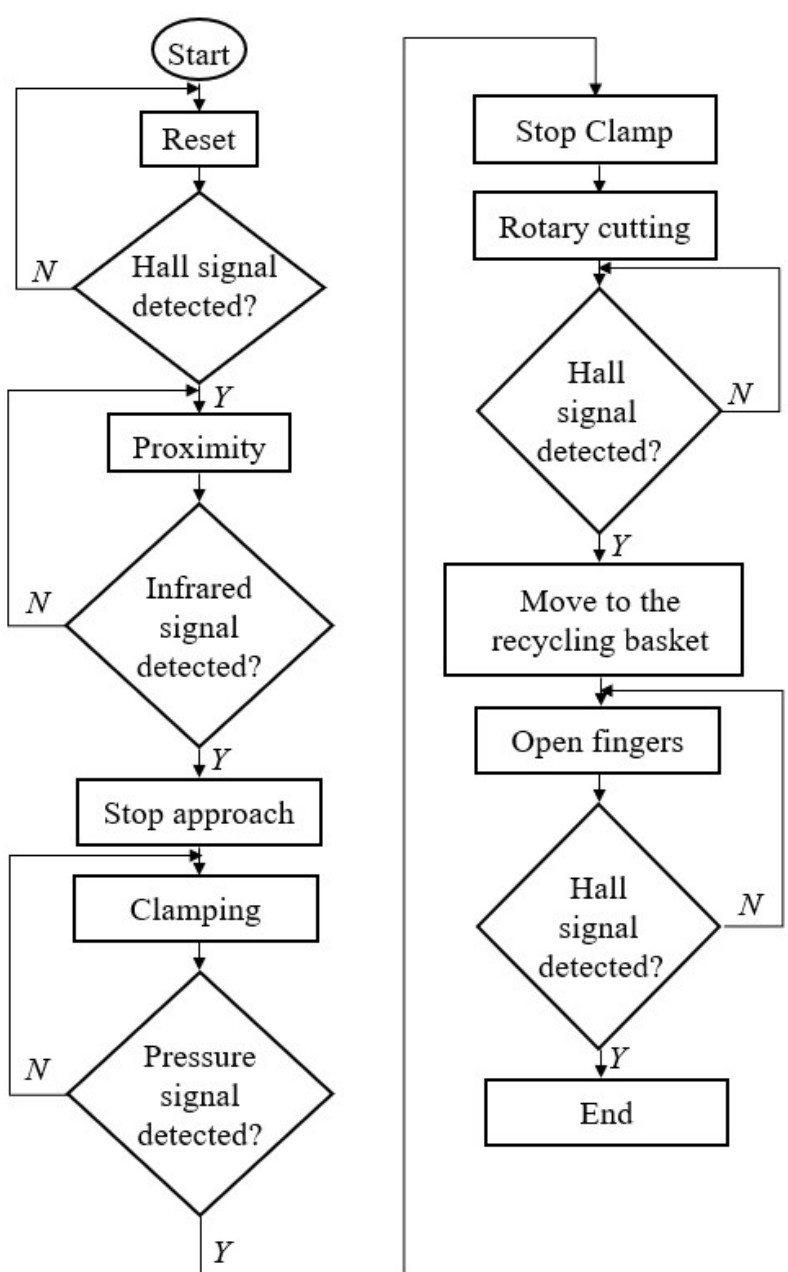

**Figure 6.** Flow chart of harvesting operation for end-effector.

*2.3. Picking Simulation Analysis*

In order to analyze the performance of the end-effector for picking citrus fruits and the relationship between the key components of the end-effector, we modeled the end-effector in PROE5.0 and imported it into ADAMS in STEP format. The end-effector was modeled in PROE5.0 and imported into ADAMS in STEP format to establish a virtual citrus picking end-effector model, and the kinetic simulation analysis of the picking method of the end-effector was carried out to verify the design of the end-effector and the feasibility and reliability of the picking method.

### 2.3.1. ADAMS-Based End-of-Pick Actuator Model

The three-dimensional virtual model of the citrus picking end-effector is imported into the ADAMS program to simulate the movement of the flexible finger of the citrus picking end-effector when gripping citrus fruits and analyze the movement and force of each part of the citrus picking end-effector in the simulated process of picking citrus fruits. The citrus

picking end-effector consists of three parts: gripping finger, cutting blade and fixed base. The gripping finger adopts the imitation fin type flexible finger, which can be driven by the stepping motor to complete the bending of the gripping finger for the opening and closing actions to complete the picking.

The speed of the stepper motor is set to 250 r/min, and the corresponding JOINT motion subsets are established for each degree of freedom in ADAMS, and the corresponding drive function is established for the JOINT, and the initial value of the function is set to 0 to prepare for the setting of the input variables. The simulation model of the citrus picking end-effector is shown in Figure 7.

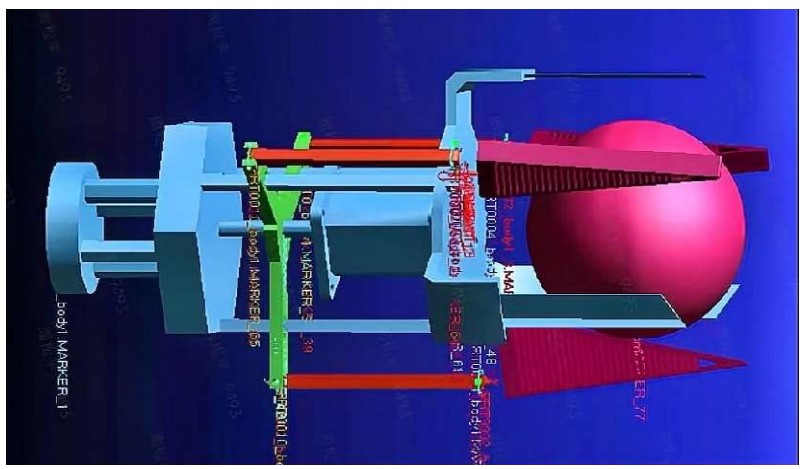

**Figure 7.** End-effector model in ADAMS.

### 2.3.2. Picking Simulation Analysis

The three fingers were evenly distributed at 120° intervals around the palm, so when the fruit was grasped in the simulation, the fruit was only roughly placed in the center of the palm, and no accurate coordinate positioning was performed, so that the robustness of the manipulator could be better verified. The torque variation curve of the flexible finger during the grasping of the citrus fruit is shown in Figure 8a, and the displacement velocity variation curve of the clamping finger is shown in Figure 8b.

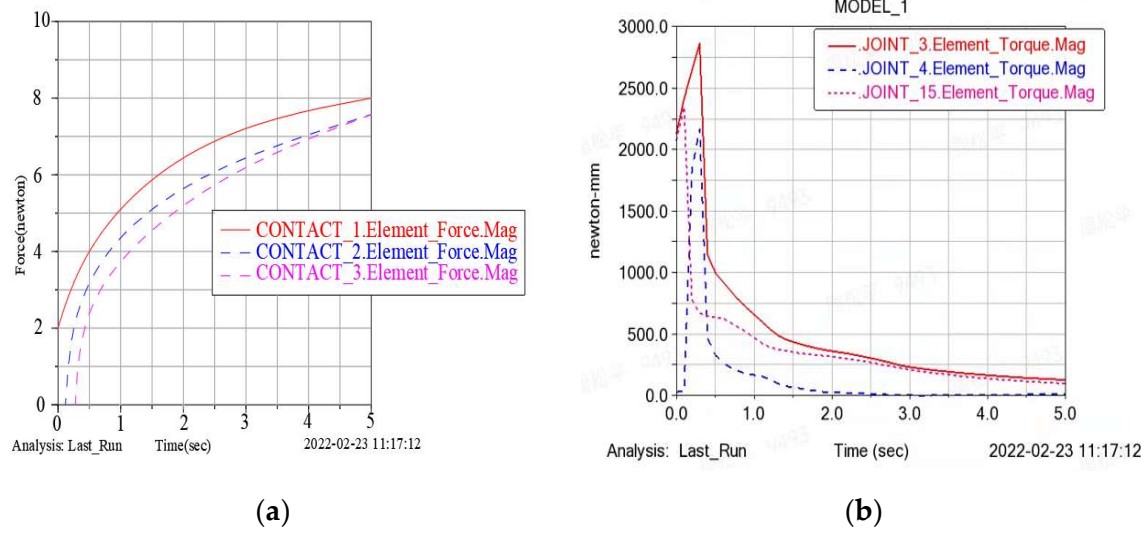

**Figure 8.** Simulation analysis diagram. (**a**) Flexible finger torque variation graph; (**b**) Clamping finger displacement velocity variation graph.

From Figure 8a, the gripping fingers of the end-effector do not cause mechanical damage to the citrus fruit when the gripping action is performed; the gripping force is between 6 and 8 N, which does not exceed the safety value of 11.03 N that the citrus fruit can withstand. Since the fruit is not placed in the center, the three fingers do not touch the fruit at the same time, so the moment value in the graph does not start to change at the same time. From Figure 8b, it can be seen that the movement of the clamping fingers is smooth, and the speed change curve is smooth when the clamping operation is carried out on the citrus fruit, which means that this scheme can successfully complete the stable grasping of the citrus fruit.

### 2.4. Fruit Stalks Cutting Simulation Analysis

### 2.4.1. Build Simulation Environment

The establishment of the simulation environment is divided into two parts: the setting of the mechanical properties to cut the fruit stalk and the selection of the cutting method. In this paper, a cylinder is used to simulate a citrus fruit stalk as the cutting object, and its diameter is set according to the diameter range of the citrus fruit stalk, with the maximum diameter set to 4.5 mm and the minimum diameter set to 1 mm. In order to observe the stress changes between the cutting blade and the stalk surface, the peak value of the cutting blade and the stalk surface was set in the simulation environment.

### 2.4.2. Parameter Settings

In order to ensure the reliability of the numerical simulation, the 3D geometric model of the cutting blade and the citrus stalk should be as realistic as possible. The length of the cutting blade is 350 mm, and the thickness of the blade is 0.8 mm; the diameter of the citrus stalk is 3 mm, and the length is 120 mm. The parameters of the virtual citrus stalk and the cutting blade in the simulation environment are set in Figure 9.

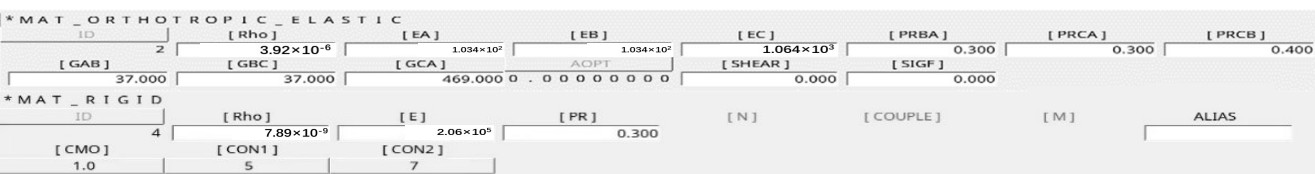

**Figure 9.** Setting of parameters for citrus fruit stalks and blade models.

### 2.4.3. Fruit Stalk Cutting Simulation Test

The end-effector must separate the citrus fruit from the fruit stalk in order to pick the citrus fruit, and the end-effector is simulated to cut the citrus fruit stalk. In the process of the cutting simulation, the end-effector is driven by the rotation of the end of the picking robot arm to cut, and the cutting process is that the end-effector blade touches the citrus fruit stalk until it is cut off.

### 2.4.4. Picking Simulation Analysis

The process of cutting the fruit stalk with the blade is shown in Figure 10. t = 0 s when the end-effector blade reaches the edge of the fruit stalk. When contact occurs between the end-effector blade and the stalk, the maximum stress is generated in the stalk at the contact position, and when the stress in the stalk exceeds the failure stress, the unit exceeding the failure stress fails, and the stalk ruptures. As the end-effector blade continues to advance, the ruptured portion of the stalk expands until the stalk is completely severed. The contact force between the end-effector blade and the fruit stalk changes, as shown in Figure 11; with the contact area between the end-effector blade and the fruit stalk increasing, the contact force increases, and when the fruit stalk is cut off, the contact force begins to decrease.

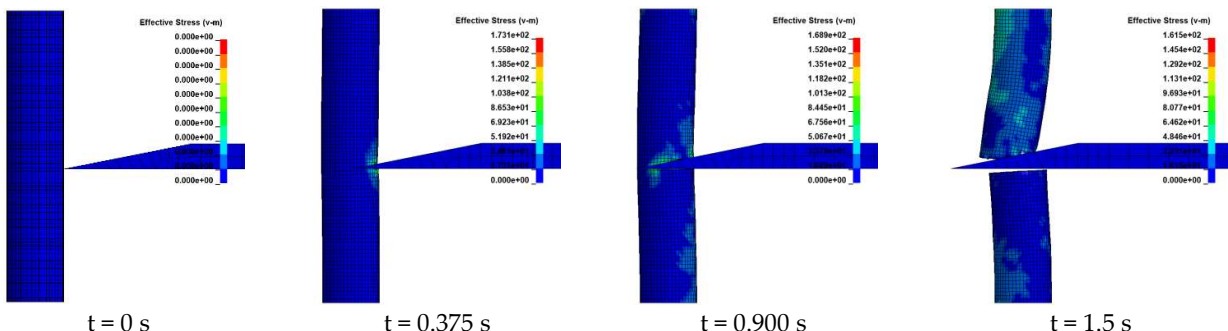

|     |     |     |     |
|-----|-----|-----|-----|
| t = 0 s | t = 0.375 s | t = 0.900 s | t = 1.5 s |

**Figure 10.** Stress diagram for citrus fruit stalk cutting simulation.

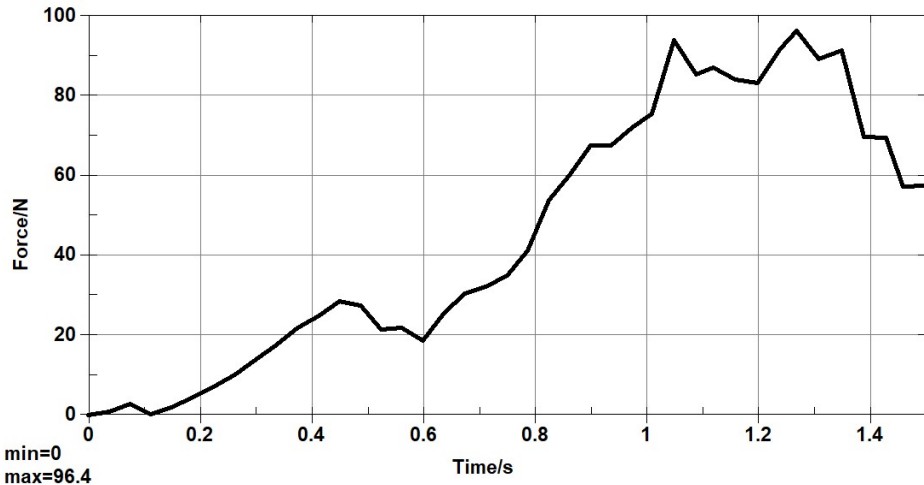

min=0
max=96.4

**Figure 11.** Simulation results graph.

## 3. Picking Performance Tests

### 3.1. Test Materials

The test subjects were citrus fruits grown in the south of China. In order to better study the physical characteristics of citrus fruits, 5 groups of citruses were randomly selected, there was 20 citruses in each group, and the dimensions of the citrus fruits were recorded, and the mean, standard deviation and coefficient of variation were calculated (Table 1).

**Table 1.** Physical characteristics of citrus fruit.

| Measured Values | Average Value/mm | Standard Deviation/mm | Coefficient of Variation |
|-----------------|------------------|-----------------------|--------------------------|
| Longitudinal diameter of citrus | 50.24 | 5.630 | 0.121 |
| Cross diameter of citrus | 52.38 | 6.125 | 0.103 |

### 3.2. Test Methods and Evaluation

The experiment was conducted using single-factor and multi-factor orthogonal tests. The end-effector was mounted on the picking robot arm for the end-effector picking performance test, as shown in Figure 12. Without considering the robot vision, the citrus fruits were placed at the center of the gripping fingers, and the effects of stepper motor speed, end-effector speed, and picking angle on the picking performance were investigated, while the picking success rate and single fruit picking time were focused as the evaluation indexes because there was no breakage in picking. A total of 105 picking tests were conducted.

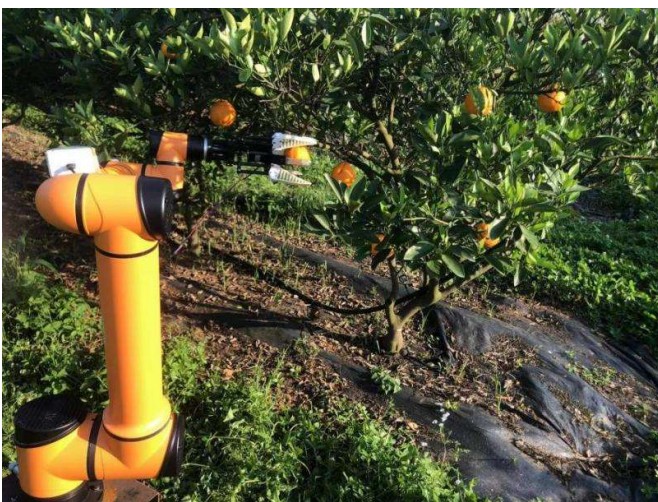

**Figure 12.** Picking test diagram of the end-effector.

### 3.3. Single-Factor Tests

There are many factors affecting the picking performance, and this study focuses on the effects of the stepper motor, end-effector speed, and picking angle on the picking performance of the end-effector with other factors held constant [50–52].

#### 3.3.1. Influence of Stepper Motor Speed on Picking Performance

The stepper motor speed controls the movement speed of this clamping mechanism. After the initial debugging of the stepper motor, it was found that when the stepper motor speed exceeded 300 r/min, the end-effector system was extremely unstable and caused damage to the fruit. Therefore, this paper selects the stepper motor speeds 200, 225, 250, 275, and 300 r/min for single-factor experimental research; analysis of the stepper motor speed changes on the impact of picking performance.

According to Figure 13 below, the picking success rate was the first to rise and then fall as the speed of stepper motor increased, and the picking success rate was the largest at 250 r/min, which could reach 95.23%; the single fruit picking time decreased as the speed of the stepper motor increased, and the single fruit picking time was the least at 300 r/min, and the picking time was 4.66 s, but in some situations, the picked citrus fruits had skin abrasion. It can be concluded that in order to ensure a better picking success rate, the speed of the stepper motor should be set to 250 r/min.

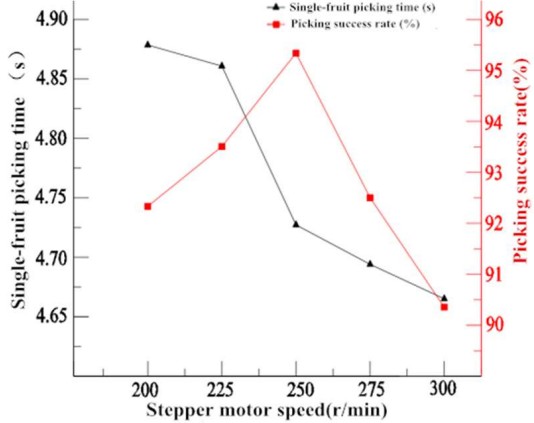

**Figure 13.** Effect of different stepper motor speeds on picking performance.

### 3.3.2. Effect of End-Effector Speed on Picking Performance

The end-effector steadily grips the citrus fruit through the flexible fingers, and then the whole end of the robot arm is controlled to rotate the end-effector as a whole to make the blade cut the citrus fruit stalk. Therefore, the rotational speed of the end-effector determines the cutting speed of the fruit stalk. Five factor levels of 80, 120, 160, 200, and 240 mm/min were selected for a single-factor experimental study to analyze the effect of the change of the rotational speed of the end-effector on the picking performance.

According to Figure 14, the picking success rate increases and then decreases as the speed of the end-effector increases, and the picking success rate is the highest at the end-effector speed of 160 mm/min, which can reach 96.19%; the single fruit picking time decreases as the speed of the end-effector increases, and it takes the least time at the end-effector speed of 240 mm/min, which is 4.65 s. At the end-effector speed of 120 mm/min, the slope of single fruit picking time decreases. In summary, it can be concluded that the picking performance is optimal at the end-effector speed of 160 mm/min.

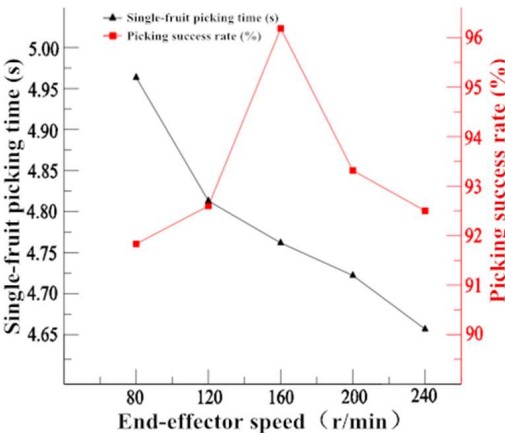

**Figure 14.** Effect of different end-effector speeds on picking performance.

### 3.3.3. Effect of Picking Angle on Picking Performance

The picking angle determines whether the end-effector can achieve complete clamping of citrus fruits and achieve shearing of the fruit stalk. In order to arrive at the optimal picking angle, the five factor levels of $-15°$, $0°$, $15°$, $30°$, and $45°$ were selected for a single-factor experimental study to analyze the effect of variation in picking angle on picking performance.

According to Figure 15, the picking success rate was the highest when the picking angle was $0°$, reaching 94.29%; the single fruit picking time was the lowest when the picking angle was $0°$, at 1.65 s. In summary, the picking performance was the best when the picking angle was kept at $0°$.

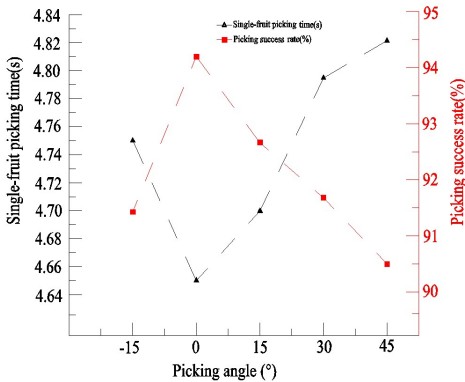

**Figure 15.** The effect of different picking angles on picking performance.

### 3.4. Multi-Factor Tests

Considering the mutual influence of each parameter, multi-factor orthogonal tests were conducted for stepper motor speed, end-effector speed, and picking angle based on the results of single-factor tests, and the results were averaged five times for each group of tests. The experimental design is shown in Table 2.

**Table 2.** End-effector picking performance test factor level coding table.

| Level | Factor | | |
| --- | --- | --- | --- |
| | $X_1$/(r/min) | $X_2$/(mm/min) | $X_3$/(°) |
| 1.682 | 200 | 80 | −15 |
| 1 | 225 | 120 | 0 |
| 0 | 250 | 160 | 15 |
| −1 | 275 | 200 | 30 |
| −1.682 | 300 | 240 | 45 |

Note: $X_1$ is the stepper motor speed; $X_2$ is the end-effector speed; $X_3$ is the picking angle.

In order to find the best combination of end-effector structures, the level factors of the orthogonal test were determined based on the comprehensive analysis of the single-factor test results, and the orthogonal test of picking performance was carried out [53]. The test results are presented in Table 3.

**Table 3.** Picking performance orthogonal test results.

| Serial Number | Level of Factor | | | Picking Success Rate Q/% | Single-Fruit Picking Time T/s |
| --- | --- | --- | --- | --- | --- |
| | $X_1$/(r/min) | $X_2$/(mm/min) | $X_3$/(°) | | |
| 1 | −1 | −1 | −1 | 92.38 | 4.95 |
| 2 | 1 | −1 | −1 | 90.05 | 4.88 |
| 3 | −1 | 1 | −1 | 93.61 | 4.65 |
| 4 | 1 | 1 | −1 | 92.64 | 4.84 |
| 5 | −1 | −1 | 1 | 91.88 | 4.54 |
| 6 | 1 | −1 | 1 | 90.38 | 4.68 |
| 7 | 1 | 1 | 1 | 91.35 | 4.62 |
| 8 | 1.682 | 0 | 0 | 93.21 | 4.85 |
| 9 | 1.682 | 0 | 0 | 94.22 | 4.74 |
| 10 | 0 | 1.682 | 0 | 92.56 | 4.68 |
| 11 | 0 | −1.682 | 0 | 91.62 | 4.69 |
| 12 | 0 | 0 | 1.682 | 91.46 | 4.66 |
| 13 | 0 | 0 | 1.682 | 91.20 | 4.82 |
| 14 | 0 | 0 | 0 | 93.25 | 4.86 |
| 15 | 0 | 0 | 0 | 92.88 | 4.69 |
| 16 | 0 | 0 | 0 | 91.95 | 4.73 |

The regression coefficients and equations between picking success rate Q, single fruit picking time T, stepper motor speed $X_1$, end-effector speed $X_2$, and picking angle $X_3$ were obtained by using Design-Expert 8.0 software (Stat-Ease, Minneapolis, MN, USA).

Picking success rate $Q$:

$$Q = 33.2 + 0.32X_1 + 1.35X_2 - 0.85X_3 - 0.85X_1X_2 + 0.15X_2X_3 - 1.62X_1X_3$$
$$+0.32X_1^2 + 0.62X_2^2 - 1.62X_3^2$$

Single-fruit picking time $T$:

$$T = 15.32 + 1.23X_1 - 2.23X_2 + 1.52X_3 + 1.62X_1X_2 - 2.33X_2X_3 + 1.82X_1X_3$$
$$-0.1X_1^2 + 2.85X_2^2 - 5.88X_3^2$$

In summary, the magnitude of factors affecting the picking success was $X_2$, $X_1$, and $X_3$, in order; the magnitude of factors affecting single fruit picking time was $X_2$, $X_1$, and $X_3$, in order.

To verify the effectiveness of the citrus picking end-effector designed in this paper, we compared the common three-finger under-driven end-effector and suction cup end-effector in the market for the test and the results of the grasping test are shown in Table 4.

**Table 4.** Picking comparison test.

| Types | Picking Number | Damage Rate | Picking Success Rate | Single-Fruit Picking Time |
|---|---|---|---|---|
| Three-finger under-driven end-effector [17] | 180 | 5.56% | 91.67% | 4.68 s |
| Suction cup end-effector [19] | 180 | 3.33% | 87.22% | 3.98 s |
| End-effector of this article | 180 | 1.11% | 93.33% | 4.05 s |

According to the comparison test results (Table 4), it can be seen that the end-effector designed in this paper has the best picking performance, with the highest picking success rate of 93.33%, the lowest damage rate of 1.11%, and a single fruit picking time of 4.05 s, which is slightly lower than that of the suction cup end-effector.

## 4. Conclusions

In order to improve the picking performance of the picking robot, according to the characteristics of citrus fruit easy to form "clumps", this paper designed high-speed citrus picking three-finger grasping end-effector that can be positioned near the center of the fruit; the device produces a human-like grasp through the precise control of the joint manual picking process motor, greatly improving the quality and efficiency of the picking part. The operating parameters of the end-effector were analyzed and optimized through finite element analysis and picking performance tests, and finally, the method was shown to be robust through tests, as summarized below.

(1) The end-effector is mainly composed of a clamping mechanism, cutting mechanism, and stepper motor. The clamping mechanism adopts the principle of three-finger clamping, which can carry out stable clamping according to the different sizes of the fruits. The cutting mechanism is fixedly installed on both sides of the clamping fingers, which can cut the stem of the fruit quickly and effectively.

(2) Through the simulation analysis of the picking process of the citrus picking end-effector, it is concluded that the clamping fingers can complete the stable picking of citrus fruits within the safe clamping force for citrus fruits, and the cutting blades can complete the cutting of citrus fruit stems stably, and the overall picking performance of the end-effector is stable.

(3) The single-factor test of the end-effector's picking performance shows that the picking performance is best when the stepper motor speed is 250 r/min, the end-effector speed is 160 mm/min, and the picking angle is 0°. Multi-factor or orthogonal tests showed that the factors affecting the end-effector picking performance were end-effector speed, stepper motor speed, and picking angle, in that order.

## 5. Patents

This research has applied for a patent: a citrus picking robot and its end-effector and citrus picking method, China, Invention Patent, ZL202110372502.7.

**Author Contributions:** Supervision, Y.W.; Visualization, Y.J.; Writing—original draft, X.X. All authors have read and agreed to the published version of the manuscript.

**Funding:** This work is partially supported by the National Natural Science Foundation of China (Grant Nos. 62027810, 61733004), the National Key Research and Development Program of China

(2020YFB1712600) and the Hunan Science and Technology Program of Hunan Province (2017XK2102, 2018GK2022). It is also supported by China Scholarship Council (CSC NO. 201806130027).

**Institutional Review Board Statement:** Not applicable.

**Informed Consent Statement:** Not applicable.

**Data Availability Statement:** Choose to ex-clude this statement.

**Conflicts of Interest:** The authors declare no conflict of interest.

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
