# Peer review of "End-Effectors Developed for Citrus and Other Spherical Crops"

_applsci, doi:10.3390/app12157945_

Round 1
Reviewer 1 Report
The main contributions of this paper are not clear to me. It is necessary to improve the results and prepare the main differences and answer the main questions, including why are these systems needed?
Author Response
Thanks to the suggestions made by the reviewers, we have improved the paper by adding a literature review on end-effectors, as well as related component design content, and finally adding a comparative picking performance test with existing end-effectors. In my opinion, the outstanding contributions of the paper are mainly: 1. the design of a three-finger end-effector based on manual picking of citrus fruits; 2. the harvesting of citrus with random stem orientation; and 3. the good picking performance in comparison with existing mechanisms.
Reviewer 2 Report
1- The method used in the article seems to be a very common robot gripper at first glance. It is sufficient to present a non-existent system, but for it to be scientific, it should be explained what the difference of this system is from the others and what kind of improvement it has made. Traditionally, 3-finger mechanisms are quite common. Why were 4 fingers used in the study? The literature review is quite weak. Soft robotic and rigid robotic gripper issues should definitely be mentioned. In the introduction part, scientific additives provided according to the literature should be presented in the form of articles. In order to obtain a patent,
2- What if the same mechanism was produced in a different way (hard, soft, complient)? What would happen if the same transaction was carried out with a system that could be considered a competitor was not investigated. Therefore, the presented article consists of introducing a system made by the authors.
3- I recommend following an approach such as progress, development and research by building on existing methods. The article is far from making a scientific contribution in its current form. It would be nice to take a simultaneous image of the robot during citrus picking and present sensor data during this time, the simulation findings are very crude in this type of study. It may be sufficient to present the simulation design phase, but different designs have not been compared at this stage. For example, instead of the preferred lengths, what would it mean to be a little longer or shorter, to have 3 fingers instead of 4, the answers to these questions could be given quite easily with simulation. Finally, if the simulation findings were compared with the actual application results, we could say that this manuscript is sufficient.
4- Figures should be presented with larger visuals. Some graphics are difficult to read. Please increase the graphic and visual quality. It would be more reader friendly if the part name was written directly instead of numbering in Figure-1.
5- In the conclusion part, it is necessary to explain the contributions obtained by mentioning why the design features are preferred.
My personal opinion about the manuscript is a major revision.
Author Response
Point 2: The probability for a given country h to be in a class k should be the proportion of observations (households) in country h that belong to the income class k. On page 9, the first equation (it would be easier for the reader if the equation is numbered) is not exactly the proportion of people because the authors take the sum of the probability. The interpretation of the equation in not obvious. Normally, after estimating a mixture of regression model we have for each observation its estimated probabilities to be classified into the different classes identified. What is often done is to classify a given observation into the class where its estimated probability is higher. In many software this is also the method used that gives us the proportion of people in each of the classes. The authors should explain the equation on page 9 and how to interpret it. Alternatively, they may use the proportion approach which will make the interpretation easier.
Response 2: We introduced the self-designed system based on the reviewers' comments, improved the description of the control system and updated the pictures and flowcharts, for example.“The stepper motor controller uses STM32F103C816, adding a 12v power supply module to provide the required power for the stepper motor. The power of the motor is relatively large, in order to protect the control circuit, drive circuit, so that the stepper motor does not burn other components in the event of a short circuit and other faults, the circuit is added to the optocoupler isolation. Add A4988 compiler in the circuit to adjust the motor's steering and motor steps. Add serial communication module to realize data transmission between control board and upper computer. Add key control in the circuit: KEY0 controls the rotation and stop of robot arm 1; KEY1 controls the positive rotation of robot arm 1; KEY2 controls the reverse rotation of robot arm 1; KEY3 controls the reverse rotation of robot arm 2; KEY4 controls the positive rotation of robot arm 2; KEY5 controls the rotation and stop of robot arm 2; SW1 reset button. When the picking operation starts, the switch is opened to provide 12V current for the stepper motor, and the current is shunted to the microcontroller for power supply when it passes through A4988, so that the microcontroller starts the stepper motor; the optocoupler isolates and uploads the signal to the stepper motor to control the lateral movement of the stepper motor. When the stepper motor moves up, it pushes the linkage mechanism to control the gripping finger for opening movement; when the stepper motor moves down, it pushes the linkage mechanism to control the gripping finger for closing movement. The schematic diagram of the control system is shown in Figure 5.
Figure 5. Schematic diagram of the control system structure
The gripping process of this end-effector system consists of two parts: finger bending and overall rotation of the end-effector, as shown in Figure 2. The actuator finger finger bending has good shape adaptiveness, when grasping larger size citrus fruit, each joint finger surface can fit in turn; when grasping smaller size citrus fruit, it can complete the parallel pinching of fingertips. And the end-effector overall rotation is based on current feedback active soft control, when grasping different elliptical citrus, according to the motor current feedback drive joint rotation, until the axial rotation force coupling eliminated, finger surface perpendicular to the cross-section tangent. The end-effector picking operation flow is shown in Figure 6.
Figure 6. Flow chart of harvesting operation for end-effector
Point 3: I recommend following an approach such as progress, development and research by building on existing methods. The article is far from making a scientific contribution in its current form. It would be nice to take a simultaneous image of the robot during citrus picking and present sensor data during this time, the simulation findings are very crude in this type of study. It may be sufficient to present the simulation design phase, but different designs have not been compared at this stage. For example, instead of the preferred lengths, what would it mean to be a little longer or shorter, to have 3 fingers instead of 4, the answers to these questions could be given quite easily with simulation. Finally, if the simulation findings were compared with the actual application results, we could say that this manuscript is sufficient.
Response 3: The author would like to thank the reviewer for the suggestion, In order to analyze the end-effector's performance on citrus fruit picking and the relationship between the key components of the end-effector, we modeled the end-effector in PROE5.0 in 3D and imported it into ADAMS in STEP format. The end-effector was modeled in PROE5.0 and imported into ADAMS in STEP format to establish a virtual citrus picking end-effector model, and the kinetic simulation analysis of the picking method of the end-effector was carried out to verify the design of the end-effector and the feasibility and reliability of the picking method. The feasibility and reliability of the end-effector design and picking method were verified.
Translated with www.DeepL.com/Translator (free version)
Point 4: Figures should be presented with larger visuals. Some graphics are difficult to read. Please increase the graphic and visual quality. It would be more reader friendly if the part name was written directly instead of numbering in Figure-1.
Response 4: Many thanks to the reviewers for their questions, we have removed the figures in Figure 1 and replaced them directly with examples.
Point 5: In the conclusion part, it is necessary to explain the contributions obtained by mentioning why the design features are preferred.
Response 5: Many thanks to the reviewers for their questions, and we followed the reviewers' suggestion to indicate in the conclusion section why the design was preferred to highlight the contribution, so we revised the conclusion as follows: In order to improve the picking performance of the picking robot, according to the characteristics of citrus fruit easy to form "clumps", this paper designed a high-speed citrus picking three-finger grasping end-effector that can be positioned by the center of the fruit, the device to produce a human-like grasp through precise control of the joint manual picking process motor, greatly improving the quality and efficiency of the picking part. The operating parameters of the end-effector were analyzed and optimized through finite element analysis and picking performance tests, and finally the method was shown to be robust through tests, as summarized below.

Reviewer 3 Report
The work is very interesting and has great social impact. The filed robotic experimental results are well presented. The mechanical design of the manipulator and simulation in ADAMS are well presented. I feel like some missing information on the sensing and collaboration with human for better on-field job. The following some suggestions to consider.
Changhui, Y. A. N. G., et al. "Research and experiment on recognition and location system for citrus picking robot in natural environment." Nongye Jixie Xuebao/Transactions of the Chinese Society of Agricultural Machinery 50.12 (2019).
Inkulu, Anil Kumar, et al. "Challenges and opportunities in human robot collaboration context of Industry 4.0-a state of the art review." Industrial Robot: the international journal of robotics research and application (2021).
Yang, C. H., et al. "Integrated detection of citrus fruits and branches using a convolutional neural network." Computers and Electronics in Agriculture 174 (2020): 105469.
Please represent figure 16 as a discrete graph than continuous.
Instead of simple regression techniques, the authors would have used some recent prediction models.
Author Response
Point 1: Changhui, Y. A. N. G., et al. "Research and experiment on recognition and location system for citrus picking robot in natural environment." Nongye Jixie Xuebao/Transactions of the Chinese Society of Agricultural Machinery 50.12 (2019).
Inkulu, Anil Kumar, et al. "Challenges and opportunities in human robot collaboration context of Industry 4.0-a state of the art review." Industrial Robot: the international journal of robotics research and application (2021).
Yang, C. H., et al. "Integrated detection of citrus fruits and branches using a convolutional neural network." Computers and Electronics in Agriculture 174 (2020): 105469.
Response 1: We appreciate it very much for this good suggestion, and we have done it according to your ideas.
Point 2: Please represent figure 16 as a discrete graph than continuous.
Response 2: We appreciate it very much for this good suggestion, and we have done it according to your ideas.
Thank you and all the reviewers for the kind advice.

Reviewer 4 Report
Comments:
Interesting topic. “The invention relates to a terminal actuator of a citrus picking robot” The authors aim to design a three-finger gripper end-effector based on the principle of manual picking of citrus.
In the present form, some issues need clarity. For improving the quality of the paper, the authors can address the following comments:
1. The title and objective and application of the robot is so limited. The reviewer suggests the authors no need to determine the number of gripping fingers and type of fruits. The robot can grip apples also and other circles fruits. But, if you consider only colour then you need to improve for other applications such as red, green.
2. In the abstract lines 12-16, “ in this study ………. and finally the end-effector is picked without considering the robot vision”. The sentence included the study objective and methodology together which make the statement unreadable. Please revise the sentence.
3. This study has no literature review on the similar robot used to picking fruits. The reviewer suggests including a table and listed sown the previous studies investigated and designed robot or machines for picking fruits.
4. Please check Figure 1 legend.
5. Figure 5 legend! Please revise and clearly describe the picture.
6. Figure 10 not clear. Figure 12 please describe.
7. Figures 11, 12 should be after description.
8. Fig. A or Figure A, please check the format line 266.
9. A comparison analysis based on efficiency and other aspects, authors can choose is required between the designed robot in this study and previous machines used for the same application.
10. The programming side of the robot is not clear in the manuscript.
Author Response
Point 1: The title and objective and application of the robot is so limited. The reviewer suggests the authors no need to determine the number of gripping fingers and type of fruits. The robot can grip apples also and other circles fruits. But, if you consider only colour then you need to improve for other applications such as red, green.
Response 1: We are grateful for the suggestion. We have added the versatility of end-effectors in the abstract, introduction, and end-effectors developed for citrus and other spherical crops. For example.“In recent years, scholars at home and abroad have conducted research on end-effectors for fruit and vegetable picking, and different end-effectors have been developed for different kinds of crops such as tomato [10-11], strawberry [12], apple [13], and kiwifruit [14]. For citrus picking, the end-effector needs to adapt to different sizes and ellipticities of citrus picking needs, and it needs to complete the separation between citrus fruits and stems [15] to achieve stable picking, which is one of the key technolo-gies for the research of end-effectors of picking robots.
For the problem of nondestructive citrus picking, most scholars have used soft materials and flexible actuators to achieve [16]. However, due to the inherent instability of the pneumatic system, the problem of fruit dropping due to unstable grasping may occur. Citrus grown in natural environment can have large differences in size and el-lipticity [17-18]. While the traditional finger-clamp end-effector mechanism is simple and can effectively grasp the fruit by clamping the fingers, the fruit stalk separation is prone to cause peel breakage of the fruit [19-20].
For example, the 10-degree-of-freedom underdriven three-finger hand [21-30] de-veloped by Gosselin's group at Laval University in Canada, which is the international leader in underdriven multifinger hand control, uses only two motors, one motor is responsible for the grasping opening and closing motion of three fingers, and the other motor completes the finger steering. Rodriguze et al. developed a 15-degree-of-freedom single-motor-driven multi-finger hand, which can achieve safe and reliable grasping without any sensors and feedback control [31]. At present, most of the multi-finger hands at home and abroad with bending and torsional complex degrees of freedom and active soft control are used for humanoid dexterous operation, integrating multiple sensors, motors and actuators, with complex control and high cost [32-35].
In order to provide a picking robot with a simple and inexpensive structure, with a certain degree of flexibility, and suitable for a variety of sizes of citrus and other spherical picking end-effector. The actuator uses a finger bending grasping mechanism to achieve different size grasping of spherical fruits such as citrus, rotates for cutting between the blade and the stalk, and finally makes a prototype for experimental veri-fication. The focus of this study is on the picking mechanism. Machine vision system is not in the scope of the study.”
Point 2: In the abstract lines 12-16, “ in this study ………. and finally the end-effector is picked without considering the robot vision”. The sentence included the study objective and methodology together which make the statement unreadable. Please revise the sentence.
Response 2: We thank the reviewer for pointing out the error, and we have revised it.
Point 3: This study has no literature review on the similar robot used to picking fruits. The reviewer suggests including a table and listed sown the previous studies investigated and designed robot or machines for picking fruits.
Response 3: We thank the reviewers for their suggestions, and we have added a comparison trial, mainly for:” To verify the effectiveness of the citrus picking end-effector designed in this paper on citrus fruit picking, we compared the common three-finger under-driven end-effector and suction cup end-effector in the market for the test, and the results of the grasping test are shown in Table 4.
Table 4. Picking comparison test
|
Types |
Picking Number |
Damage rate |
Picking success rate |
Single-fruit picking time |
|
three-finger under-driven end-effector |
180 |
5.56% |
91.67% |
4.68s |
|
suction cup end-effector |
180 |
3.33% |
87.22% |
3.98s |
|
End-effector for this article |
180 |
1.11% |
93.33% |
4.05s |
According to the comparison test results (Table 4), it can be seen that the end-effector designed in this paper has better picking performance, with the highest picking success rate of 93.33%, the lowest damage rate of 1.11%, and a single fruit picking time of 4.05s, which is slightly lower than that of the suction cup end-effector.”
Point 4:Please check Figure 1 legend.
Response 4: We thank the reviewers for their suggestions and we have made the requested changes to Figure 1.
Point 5: Figure 5 legend! Please revise and clearly describe the picture.
Response 5: We thank the reviewers for their suggestions and we have revised Figure 5 as requested and added a description of the self-designed control system.
Figure 5. Schematic diagram of the control system structure
The gripping process of this end-effector system consists of two parts: finger bending and overall rotation of the end-effector, as shown in Figure 2. The actuator finger finger bending has good shape adaptiveness, when grasping larger size citrus fruit, each joint finger surface can fit in turn; when grasping smaller size citrus fruit, it can complete the parallel pinching of fingertips. And the end-effector overall rotation is based on current feedback active soft control, when grasping different elliptical citrus, according to the motor current feedback drive joint rotation, until the axial rotation force coupling eliminated, finger surface perpendicular to the cross-section tangent. The end-effector picking operation flow is shown in Figure 6.
Figure 6. Flow chart of harvesting operation for end-effector
Point 6: Figure 10 not clear. Figure 12 please describe.
Response 6: We thank the reviewers for their suggestions, and we have revised Figures 10 and 12 as requested.
Point 7: Figures 11, 12 should be after description.
Response 7: Thanks to the reviewer's suggestion, we have swapped the position of Figure 11 and Figure 12 and the description as requested.
Point 8: Fig. A or Figure A, please check the format line 266.
Response 8: We thank the reviewers for their suggestions, and we have made the requested changes.
Point 9: A comparison analysis based on efficiency and other aspects, authors can choose is required between the designed robot in this study and previous machines used for the same application.
Response 9: Thanks to the reviewer's suggestion, we have added a comparison picking trial after the picking trial.
Point 10: The programming side of the robot is not clear in the manuscript.
Response 10: Thanks to the reviewer's suggestion, we have improved the control system design in the paper, considering that the programming aspect is very simple with serial communication, do you think this is feasible?

Round 2
Reviewer 1 Report
the revision made by the authors. but, as i checked the submitted files, there is no response to my comments. in this regard, it should be resubmitted.
Author Response
Response 1: Thanks to the reviewer's suggestion, we have improved the whole text. We mainly designed a three-finger citrus picking end-effector to realize mechanization and intelligence of citrus picking. The actuator is designed to achieve stable picking of citrus of different sizes and ellipticality by fully grasping the citrus with three fingers and then rotating the citrus as a whole to complete the separation of citrus fruit and fruit stalk. Then, the effectiveness of the end-effector in picking citrus fruits was demonstrated by simulating the end-effector in the picking process. Finally, a picking test was conducted without considering robot vision. The test results show that the end-effector has a picking success rate of 95.23% for citrus with a diameter of 30-100 mm and an average picking time of 4.65 seconds for a single fruit. This end-effector can realize the picking function for different sizes and shapes of citrus, and has the advantages of high adaptability, stable gripping and no damage to the fruit.In my opinion, the outstanding contributions of the paper are mainly: 1. the design of a three-finger end-effector based on manual picking of citrus fruits;
“The end-effector of the picking robot holds the citrus fruits by the clamping fingers. To adapt to the different sizes of citrus fruits, the clamping fingers are driven by the stepper motor to maximize the opening of the clamping fingers before grabbing the fruits. When the end-effector reaches the picking point, the gripping finger slides downward under the positive drive of the stepper motor, thus driving the gripping finger to close the movement until the gripping finger makes adaptive contact with the target fruit to form a certain envelope on the surface of the citrus, generating a large friction force with a small gripping force to reliably grab the citrus fruit without causing damage to the citrus fruit. After the gripping fingers grip the citrus fruit, the end-effector rotates as a whole, so that the cutting blade fixed between the gripping fingers finishes cutting the citrus fruit stalk.”
- the harvesting of citrus with random stem orientation;
- the good picking performance in comparison with existing mechanisms.
To verify the effectiveness of the citrus picking end-effector designed in this paper on citrus fruit picking, we compared the common three-finger under-driven end-effector and suction cup end-effector in the market for the test, and the results of the grasping test are shown in Table 4.
Table 4. Picking comparison test
|
Types |
Picking Number |
Damage rate |
Picking success rate |
Single-fruit picking time |
|
three-finger under-driven end-effector[54] |
180 |
5.56% |
91.67% |
4.68s |
|
suction cup end-effector[55] |
180 |
3.33% |
87.22% |
3.98s |
|
End-effector for this article |
180 |
1.11% |
93.33% |
4.05s |
According to the comparison test results (Table 4), it can be seen that the end-effector designed in this paper has better picking performance, with the highest picking success rate of 93.33%, the lowest damage rate of 1.11%, and a single fruit picking time of 4.05s, which is slightly lower than that of the suction cup end-effector.

Reviewer 2 Report
Accept in present form
Author Response
It is a great honor to have your approval of our work.
Reviewer 4 Report
The authors make most of the requested reviewer concerns. However there are some need to be clarify:
1. Point 2, the statement not received from previous comment.
2. Point 3 comparison table added but there is no references on the table.
3. Point 8 fig.9 still format still not fixed.
4. point 9 in previous comments has no response in the manuscript.
Author Response
Point 1: Point 2, the statement not received from previous comment.
” In the abstract lines 12-16, “ in this study ………. and finally the end-effector is picked without considering the robot vision”. The sentence included the study objective and methodology together which make the statement unreadable. Please revise the sentence.”
Response 1: We are grateful for the suggestion. We made changes based on the reviewers' suggestions and replaced the previously incorrect statement with “Finally, a picking test was conducted without considering robot vision.”.
Point 2: Point 3 comparison table added but there is no references on the table.
Response 2: Many thanks to the reviewers for their suggestions and we have added it to my paper based on the references we have reviewed previously.
Table 4. Picking comparison test
|
Types |
Picking Number |
Damage rate |
Picking success rate |
Single-fruit picking time |
|
three-finger under-driven end-effector[54] |
180 |
5.56% |
91.67% |
4.68s |
|
suction cup end-effector[55] |
180 |
3.33% |
87.22% |
3.98s |
|
End-effector for this article |
180 |
1.11% |
93.33% |
4.05s |
- Fan P, Yan B, Wang M, et al. Three-finger grasp planning and experimental analysis of picking patterns for robotic apple harvesting[J]. Computers and Electronics in Agriculture, 2021, 188: 106353.
- Kurpaska S, Sobol Z, Pedryc N, et al. Analysis of the pneumatic system parameters of the suction cup integrated with the head for harvesting strawberry fruit[J]. Sensors, 2020, 20(16): 4389.
Point 3: Point 8 fig.9 still format still not fixed.
Response 3: We thank the reviewers for their careful review, and we have revised the manuscript according to the errors pointed out by the reviewers.“From Figure 8a……Figure 8b,”
Point 4: point 9 in previous comments has no response in the manuscript.
“A comparison analysis based on efficiency and other aspects, authors can choose is required between the designed robot in this study and previous machines used for the same application.
Response 4: Thanks to the suggestions made by the reviewers, we have compared the picking performance of our self-designed end-effector with commercially available mechanisms in a test, and the results of the comparison show that he end-effector designed in this paper has better picking performance, with the highest picking success rate of 93.33%, the lowest damage rate of 1.11%, and a single fruit picking time of 4.05s, which is slightly lower than that of the suction cup end-effector.